# Central Nervous System Effects of Oral Propranolol for Infantile Hemangioma: A Systematic Review and Meta-Analysis

**DOI:** 10.3390/jcm8020268

**Published:** 2019-02-22

**Authors:** Thuy Thai, Ching-Yu Wang, Ching-Yuan Chang, Joshua D. Brown

**Affiliations:** Department of Pharmaceutical Outcomes and Policy, College of Pharmacy, University of Florida, Gainesville, FL 32610, USA; thuythai@ufl.edu (T.T.); chingyuwang@ufl.edu (C.-Y.W.); c.chang@ufl.edu (C.-Y.C.)

**Keywords:** oral propranolol, infantile hemangioma, central nervous system, sleep-related event

## Abstract

Concerns about the effects of propranolol on the central nervous system (CNS) in the infantile hemangioma (IH) population have been raised. We conducted a meta-analysis of the CNS and sleep-related effects of oral propranolol in IH patients. PubMed, Embase, Cochrance, Web of Science, and Clinicaltrials.gov were searched for relevant studies. We included clinical trials that compared oral propranolol with other treatments among IH patients under 6 years old and monitored and reported any adverse events. Study characteristics, types and number of adverse events were abstracted. Cochrane Collaboration Risk of Bias Tool was used to assess risk of bias. Our main outcomes were CNS and sleep-related effects. Random-effects models were used to estimate the pooled risk ratio. We did not observe statistically significant associations between oral propranolol and CNS or sleep-related effects. Oral propranolol appeared to have a safer profile of CNS effects than corticosteroids (RR = 0.27, 95% CI 0.02–3.00), but had an increased risk versus non-corticosteroids (for CNS effect, RR = 1.40, 95% CI 0.86–2.27; for sleep-related effects, RR = 1.63, 95% CI 0.88–3.03). Despite no statistically significant associations, there were suggestive findings of increased CNS effects and sleep-related risk of propranolol versus non-corticosteroids. In practice, CNS and sleep-related events should be monitored more closely among IH patients treated with oral propranolol.

## 1. Introduction

Infantile hemangioma (IH), a form of soft-tissue tumors, is the most common vascular tumor in infants. The incidence of IH is around 4.5%, but varies from 0.2% to 10% due to differences in populations and study designs [1]. IH usually appears at birth or soon after with rapid growth during the first 3 months, achieves maximal size after 9 months, and then regresses completely after 4 years [2]. Although most IH cases are benign, 10 to 15% require treatment due to serious complications such as obstruction, ulceration, or disfigurement [2]. Treatment options for IH include systemic treatments, local treatments, and laser and surgical therapy [2]. Systemic treatments include oral propranolol and corticosteroids [2]. Since 2008, oral propranolol has replaced corticosteroids as the first line systemic treatment for IH as an off-label indication, and later on, guidelines have recommended oral propranolol for complicated IH in many countries [3,4,5,6,7]. Oral propranolol has demonstrated superior efficacy and safety relative to corticosteroids, surgery, or placebo [8,9]. It is recommended to initiate oral propranolol at age of 5 weeks to 5 months with 6.5-month treatment duration at a starting dose of 0.6 mg/kg twice daily and a maintenance dose of 1.7 mg/kg twice daily [7]. 

Concerns about the effects of propranolol on the central nervous system (CNS) in the pediatric population have been raised due to the lipophilic characteristic of propranolol which makes it easier to cross the blood–brain barrier [10]. Sleep disorders (i.e., fatigue, insomnia, nightmares, night restlessness, and sleep disturbances) and agitation were the most common CNS adverse events of propranolol in infants with IH in previous studies [9,11]. 

Previous studies have raised concerns about the safety issues regarding CNS effects of oral propranolol for IH treatment; however, the exact magnitude of CNS effects on propranolol use is unknown among infants with IH [10]. In an effort to quantify the CNS effect of propranolol, we conducted a meta-analysis to study the risk of CNS effects of oral propranolol in IH treatment as compared to a comparison group.

## 2. Experimental Section

### 2.1. Data Sources and Searches

Pubmed, Embase, Cochrane, Web of Science, and Clinicaltrials.gov were searched for relevant studies published before the searching date 1 April 2018. Detailed search terms for each database are shown in Table A1. In brief, search terms included “infantile hemangioma” and “oral propranolol”. We conducted a search strategy through a two-step approach. First, we applied the search terms for each database. After removing duplications of the search results from the five databases, we classified the studies into either review articles or other types of study. Second, for review articles, we did a hand-search to include all studies in the reference list. Then, we combined non-review studies found in the first step and the hand-search studies in the second step. After duplicate removal, a list of candidate studies was created. These studies were further considered for inclusion and exclusion based on the criteria and process described in the next section. No language, study’s location and publication date restrictions were applied. 

### 2.2. Study Selection

The candidate studies were independently screened by titles and abstracts by two reviewers. The full texts of selected studies were then retrieved and reviewed. Eligible studies were selected based on the following inclusion and exclusion criteria. We included studies that (1) were randomized controlled trials; (2) compared propranolol with other treatments (including placebo but cannot be a combination of propranolol and other treatments); (3) included participants aged under 6 years old with infantile hemangioma; and (4) reported adverse events. We excluded cohort studies, case control studies, case reports, case series, in vitro studies, narrative reviews, editorials, letters, and erratum. We contacted the authors of the original studies if the full-text was unavailable online or we needed further clarification on the study details that were not clearly stated.

### 2.3. Data Extraction and Quality Assessment

Two independent reviewers, working in duplicate, extracted the study characteristics and the reported adverse effects and assessed the risk of bias for each study A consensus discussion resolved any disagreement between them. The following information was collected: sample size, inclusion and exclusion criteria, mean or median age, propranolol treatment dosage and frequency, propranolol treatment duration, follow-up time, comparison group(s), study year(s) and country. 

We extracted all reported adverse effects from the included studies. The primary outcome of interest was CNS-related adverse effects. “Growth disability”, “growth and development”, “drowsiness”, “somnolence”, “lethargy”, “sleep disturbance or disorder” and “agitation” were considered as CNS-related adverse effects. A secondary outcome was only sleep-related adverse effects including “somnolence”, “lethargy”, “sleep disturbance” and “sleep disorder”.

Cochrane Collaboration Risk of Bias Tool was used to assess risk of bias focusing on random sequence generation, allocation concealment, blinding of participants, blinding of outcome assessment and selective reporting [12]. All the above steps were conducted independently by T.N.T. and C.Y.W., and the articles in Chinese were reviewed by C.Y.W. and C.Y.C. Discrepancies between two reviewers were reconciled through discussion. When the final decision could not be made, a third reviewer was added to reconcile the difference.

### 2.4. Data Synthesis and Analysis

Random-effects meta-analysis was performed to obtain the pooled estimate of incidence and risk ratio (RR) of CNS effects and sleep-related adverse effect. When an adverse effect was monitored but no case occurred, we estimated the risk ratios through inverse variance with 0.5 continuity correction [13]. Heterogeneity between studies was evaluated using *I*^2^, however, when the number of included studies was small (i.e., less than 10), *I*^2^ can be biased [14]. Thus, we further evaluated heterogeneity through visual assessment. A funnel plot was used to assess potential publication bias.

We conducted two sets of sensitivity analyses. First, due to the concerns for potential bias toward the null associated with the inverse variance with the 0.5 continuity correction method, several other correction methods were utilized to adjust for single-zero (in which no event was observed in only one treatment group) or double-zero (in which no event was observed in both treatment groups) studies [10]. These correction methods included inverse variance with “treatment-arm” continuity correction, Mantel-Haenszel (MH), and beta-binomial with correlated responses [10]. The pooled risk ratio derived from implementing these methods was compared to the risk ratio of the main analysis using the inverse variance with 0.5 continuity correction method. Second, due to the fact that different comparison groups (e.g., corticosteroids, laser, or placebo) might have different safety profiles, we stratified the pooled RRs by comparison groups. The correction methods mentioned in the first set of sensitivity analyses were also performed and compared in the stratified analyses.

All statistical analyses were performed using STATA version 15.1 and R version 3.4.4 with the “meta” and “mmeta” packages [15,16]. 

## 3. Results

### 3.1. Study Selection and Characteristics

Our search identified 884 relevant citations. After duplicate removal, title and abstract screening, and full-text review, 11 studies (829 participants) met the inclusion and exclusion criteria (Figure 1) [17,18,19,20,21,22,23,24,25,26,27]. The comparison treatments include prednisolone (3 studies), prednisone (1 study), atenolol (1 study), timolol (1 study), laser treatment (2 studies), and placebo (3 studies). The mean age of participants varied from 3 to 18 months. Dosages of propranolol varied from 0.5 mg/kg/day to 3 mg/kg/day. Treatment duration for propranolol ranged from 1 month to a year. Most studies did not follow patients after treatment discontinuation, except four studies (Kim et al., Malik et al., Gong et al., and Leaute-Labreze et al.) [18,20,21,24]. Inclusion and exclusion criteria varied among studies. For example, six studies only included infants without treatment history, four studies did not specify treatment history, and one study enrolled infants who failed previous corticosterioid treatment. Detailed characteristics of the included studies are listed in Table 1.

### 3.2. Study Quality Assessment and Publication Bias Evaluation

Figure 2 shows the quality assessment of the included studies. Potential biases of random sequence generation and allocation concealment were at low risk for the majority of the included studies (i.e., only several studies had unclear risk of bias, but no study was at high risk of those biases). However, five out of 11 studies did not perform adequate blinding, and three studies had high risk of attribution bias. Furthermore, all included studies had low risk of selective reporting bias or other biases. Therefore, the overall risk of bias for the included studies was low, except identified inadequate blinding and attribution biases. In terms of publication bias, the funnel plot had a symmetric inverted funnel shape (Appendix A, Figure A1), which indicates minor to no publication bias. 

### 3.3. Pooled Effect Size 

Table 2 presents details about the type and number of the CNS adverse effects for the included studies. The pooled risk ratio for CNS adverse effects of propranolol compared with other treatments is described in Figure 3. The pooled incidence of CNS effects was 7% in both propranolol and comparison groups. For sleep disorder, pooled incidence was 10% (95% CI, 3.2–27.5%) in the propranolol group and 8% (95% CI, 4.3–15.8%) in the comparison group. In the main analysis, we observed an increased, though not statistically significant, risk of CNS effects and sleep-related adverse effects. The pooled RR was 1.16 (95%CI, 0.64–2.12) for CNS effect and 1.67 (95%CI, 0.91–3.07) for sleep-related effect. Through visual assessment and *I*^2^, there was little to no heterogeneity observed in effect sizes (*I*^2^ = 18% for CNS effect and *I*^2^ = 0% for sleep-related effect) (Figure 3). 

In the first set of sensitivity analyses, the pooled RR remained similar across different approaches of adjusting for single-zero or double-zero studies (Table A2). In the second set of sensitivity analyses, four studies [18,20,22,25] that compared propranolol to corticosteroids were included which gave a risk ratio for CNS effect of 0.27 (95%CI 0.03–2.81). Though not statistically significant, this indicates that patients who received propranolol had a lower risk of CNS effects compared to those who received corticosteroids. Among the four included studies, only one reported sleep-related effects [20]; therefore, we were unable to calculate a pooled RR for the secondary outcome. Among studies with a comparison group of placebo or non-corticosteroids, despite no significant findings, we observed an increased risk in both CNS and sleep-related effects (Table A2 and Figure A2).

## 4. Discussion

This systematic review and meta-analysis provided quantitative evidence about CNS and sleep-related adverse events of oral propranolol in IH-treated patients as compared to other treatment options. Although propranolol did not have statistically significant associations with overall CNS or sleep-related effects among infants with IH, oral propranolol appeared to have a safer profile for CNS effects when compared to corticosteroids, but have an increased risk of CNS effects and sleep-related effects as compared to placebo and other non-corticosteroids comparisons. 

Previous studies have raised concerns regarding propranolol use for IH treatment during the susceptible developmental period of infancy [9,10,11]. Sleep-related events were the most common adverse events in a prior systematic review [9]. Additionally, a systematic review conducted by Léauté-Labrèze et al. found that sleep disturbances, peripheral coldness, and agitation were the most frequently reported adverse events among 5862 propranolol-treated infants in 85 identified studies. In these studies, adverse events in the propranolol group were fully captured, however safety outcomes were not captured in the comparison group [9,11]. Therefore, we restricted our research to studies that reported adverse events in both treatment and comparison groups and provided a direct comparison for CNS effects and sleep-related events between propranolol and controls. Despite no statistically significant results, our findings were consistent with point estimates that substantiate previous concerns about CNS effects and sleep-related adverse events of propranolol. 

Compared with previous studies, the current meta-analysis addressed the concern for the CNS effects of propranolol in infants with IH with two scenarios. While our study suggested that propranolol was safer than corticosteroids for CNS effects and sleep-related adverse events, the results also indicated a novel finding of an increased risk as compared to non-corticosteroids. Again, although these associations were not statistically significant, oral propranolol should be used after carefully weighing the risk and benefit. Clinicians should closely monitor CNS effects and sleep-related events in IH patients treated with oral propranolol. Moreover, because only clinical trials with short-term follow-up were included, these results are limited and the potential of long-term side effects on infant patients should be considered with further studies about long-term CNS-related effects warranted.

This study has several limitations. It provides the magnitude of CNS effects with propranolol use in the early developmental stages and within the treatment period. This information contributes to understanding of the safety profile of oral propranolol in pediatric patients. However, it is difficult to confirm the true incidence of CNS effects if a study did not specifically monitor for these events. Furthermore, CNS effects might be unnoticed and underreported in the short follow-up periods in most clinical studies. Moreover, because the number of studies and overall count of adverse outcome reported were small, our estimated RR included very wide confidence intervals. There may also be a delayed effect between propranolol use and CNS effects, which would not be captured in the short-term clinical trials that met our inclusion criteria.

## 5. Conclusions

Despite no statistically significant associations between propranolol and CNS effects or sleep-related adverse events, propranolol appeared to be safer than corticosteroids. However, there were findings suggesting an increased risk of CNS effects and sleep-related events with propranolol as compared to non-corticosteroids.

## Figures and Tables

**Figure 1 jcm-08-00268-f001:**
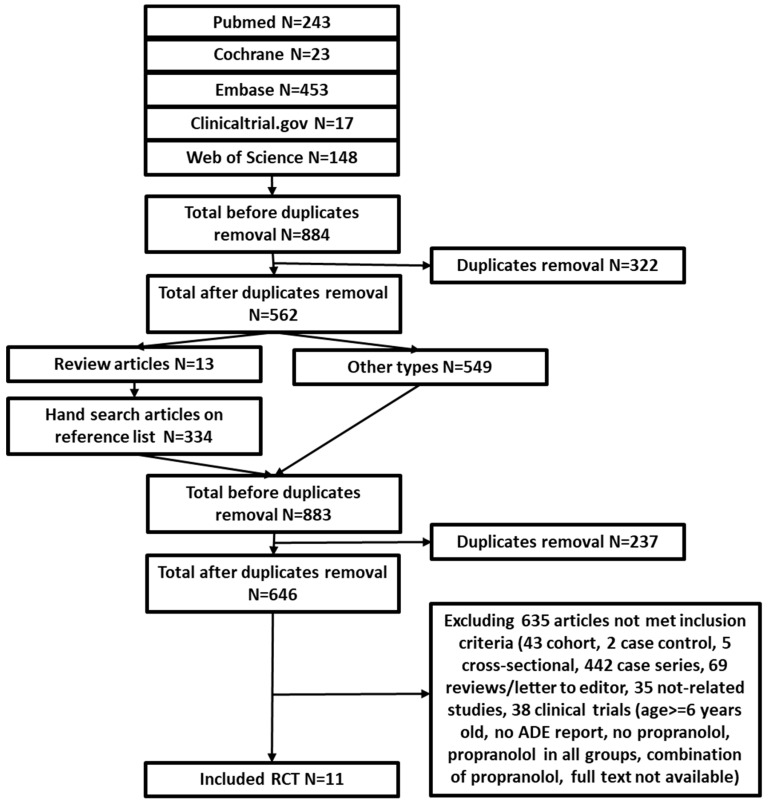
PRISMA diagram for study selection.

**Figure 2 jcm-08-00268-f002:**
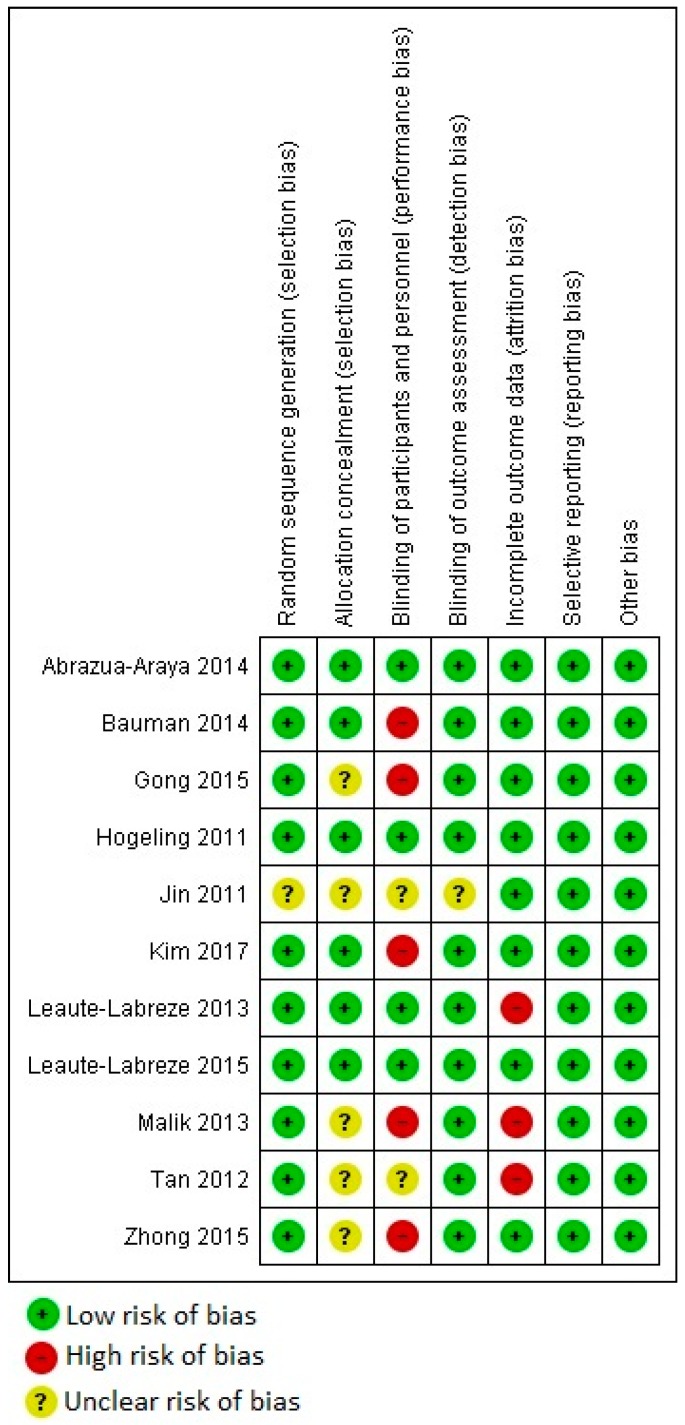
Quality assessment of the included studies.

**Figure 3 jcm-08-00268-f003:**
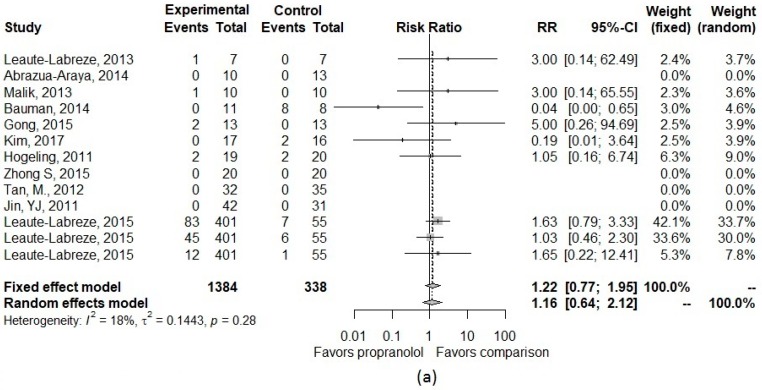
Forest plot of the pooled risk ratio of (**a**) all CNS effects and (**b**) only sleep-related effects in the main analysis.

**Table 1 jcm-08-00268-t001:** Study characteristics of included studies.

Author (Year)	Sample Size	Inclusion Criteria	Exclusion Criteria	Comparison Treatment	Age	Propranolol Treatment Dosage	Propranolol Treatment Duration	Follow-Up Time	Country	Year of Data Collection
Abrazua-Araya (2014)	23	IH needing treatment	History of allergy, hypersensitivity and treatment, heart or respiratory disease	Atenolol 1 mg/kg/d for 6 m	5.2 ± 3.5 m	2 mg/kg/d	6 m	0 m	Chile	2012–2013
Jin, YJ (2011)	73	Diagnosed with IH that influence their appearance were included.	Heart or respiratory disease	Prednisone 3 mg/kg/d, 6 m max	NR	2 mg/kg/d	6 m max	0 m	China	2009–2010
Kim, K. H. (2017)	34	IH had 10–20% volume increase in 2–4 w or IH-related dysfunction or aesthetic problem	Without normal heart function or having treatment history	Prednisolone 2 mg/kg/d	3.3 m	2 mg/kg/d	16 w	4 w	Korea	2013–2014
Leaute-Labreze, C. (2013)	14	≥ 1 nonthreatening IH > 1 cm; without vital or functional impairment; not justifying oral corticosteroids	Requiring urgent IH treatment; contraindications or history of treatment	Placebo	12 w	3–4 mg/kg/d	1 m	0 m	France	2008–2010
Malik, M. A. (2013)	20	Problematic IH	Presence or history of heart, bronchoobstructive, metabolic, or liver disease, visceral lesions, prematurity	Prednisolone 1 mg/kg/d	4–5 m	1–3 mg/kg/d	1 y max	6 m	India	2011–2012
Gong, H. (2015)	26	Superficial hemangiomas, no previous treatment	Deep/mixed haemangiomas, respiratory or heart disease, fever, diarrhea	0.5% timolol maleate eye drops	NR	1.5 mg/kg/d	5.3 m	3–12 m	China	2012–2013
Zhong S (2015)	40	Mixed or deeper IH >8 mm diameter, treatment naïve; complete treatment and follow-up	Heart and respiratory disease	Laser	3.69 m	1.5 mg/kg/d	6 m	0 m	China	2013–2014
Bauman (2014)	19	Proliferating and symptomatic IH	Inadequate social support, received other IH treatment for IH, having a co-morbidity	Prednisolone 2 mg/kg/d	2.5–4 m	2 mg/kg/d	323 d average	0 m	US	2010–2012
Hogeling, M. (2011)	39	IHs with a deep component, impair function, or aesthetic disfigurement, late or failed to respond to corticosteroid therapy.	Requiring urgent treatment, contraindications to propranolol, extracutaneous IH	Placebo	67–71 w	1–2 mg/kg/d	6.5 m	0 m	Australia	2009–2010
Leaute-Labreze, C. (2015)	456	A proliferating IH required systematic therapy	Patients with life-threatening, function-threatening, or severely ulcerated hemangiomas	Placebo	103.8 d	1–3 mg/kg/d	3 or 6 m	72 w	Multiple	2010–2011
Tan, M. (2012)	97	IH diagnosis, treatment naïve	Respiratory, cardiovascular diseases, other systematic chronic diseases	Laser	NR	0.5–1 mg/kg/d	6 m	0 m	China	2010–2011

**Table 2 jcm-08-00268-t002:** Types and number of CNS adverse effects for the included studies.

Number	Author, Year	Total Sample Size	Propranolol (n)	Comparison (n)	Type of CNS Effect	Number Patients with CNS Effect in	
						Propranolol Group	Control Group
1	Abrazua-Araya, 2014	25	10	13	Adverse event related to CNS	0	0
2	Kim, 2017	35	17	16	Growth disability	0	2
3	Leaute-Labreze, 2013	16	7	7	Drowsiness	1	0
4	Malik, 2013	22	10	10	Somnolence	1	0
5	Gong, 2015	28	13	13	Lethargy	2	0
6	Bauman, 2014	21	11	8	Growth and development	0	8
7	Hogeling, 2011	39	19	20	Sleep disturbances	2	2
8	Leaute-Labreze, 2015	456	401	55	Sleep disorder	83	7
		456	401	55	Agitation	45	6
		456	401	55	Somnolence	12	1
9	Tan, M., 2012	69	33	36	Adverse event related to CNS	0	0
10	Zhong S, 2015	42	21	21	Adverse event related to CNS	0	0
11	Jin, YJ, 2011	75	43	32	Adverse event related to CNS	0	0

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
