# Peer review of "Central Nervous System Effects of Oral Propranolol for Infantile Hemangioma: A Systematic Review and Meta-Analysis"

_jcm, 2019, doi:10.3390/jcm8020268_

Round 1
Reviewer 1 Report
I have no further comments with the exception of one minor correction:
Clinicaltrial.gov should be to Clinicaltrials.gov (with an "s") throughout the manuscript.
Author Response
We have made this one edit, thank you for catching.
Reviewer 2 Report
The authors have well summarized and analyzed the Central Nervous System and sleep-related effects of the oral propranolol in Infantile Hemangioma (HI) patients.
Overall, the review is well written and significant editing has been already made.
I have only a minor suggestion on the layout of the tables.
Would it be possible to decrease the font of the text, less writing, use abbreviations, such as 6m max (instead of 6 months maximum), dates like Oct instead of October and so on (adding major info in the figure legend below) in order to fit the table 1 in max 2 pages?
The same for other tables.
Author Response
We have reduced Table 1 to 2 pages. Thank you.
This manuscript is a resubmission of an earlier submission. The following is a list of the peer review reports and author responses from that submission.
Round 1
Reviewer 1 Report
The article presented is of great interest. However, the article does not present any new idea when compared to previous publications in the area of interest. Also, the current work misses some critical indicators for inclusion and exclusion criteria.
1) The study does not factor in previous treatments for the same disease.
2) No information was presented to conclude the preferred dosage of propranolol.
3) Also, how does study fares when compared to other beta blockers such as atenolol and timolol.
Reviewer 2 Report
In this manuscript, the authors present a systematic review and meta-analysis of the CNS effects of oral propranolol in children with an infantile hemangioma. The topic is interesting especially given the incidence of IH is ~5% of all infants and therefore may impact a significant number of children with this condition. I have several comments for the authors to consider.
It may be helpful to include the estimates of the summary RRs in the abstract.
The Introduction states that propranolol and corticosteroids are the systemic treatments for IH, but several of the studies included in the meta-analysis are RCTs comparing propranolol with non-corticosteroids – is IH typically treated with non-corticosteroid?
The authors state that propranolol was approved for treating IH by the FDA – is this drug approved for treating IH in other countries including those countries where the evaluated studies were conducted?
Please include further details about the search parameters for eligible studies, including what time period was considered (i.e., what is the latest date considered in the search), and were studies from all regions considered.
Which conditions are considered to be “CNS effects”? Were autism and seizures included in the outcome definition?
Were any cohort and case-control studies identified in the search? What is the rationale for excluding cohort and case-control studies?
In the PRISMA diagram, the authors may consider specifying the number of articles excluded based on each of the defined inclusion/exclusion criteria.
Section 3.2 is confusing as written and it is suggested that Section 3.2 be revised for clarity. Figure 2 is helpful in understanding the quality assessment of the studies included in the meta-analysis, but a legend or description of the colors/symbols is needed.
This is a systematic review, however, the studies included are not described in the main body of the article. If the authors intend for this to be a systematic review, it is suggested that there be a description of the included studies in the text of the article, and Table A2 should be moved from the appendix to the main body of the article.
Please explain what a meant by a single-zero and double-zero study.
The forest plots in Figure 3: the columns on the left side of each plot are confusing (I’m not sure what each column is showing since the headings seem to be shifted); the far-right column of each plot seems to be cut off.
Table 2: Since the results from each analysis approach seem to be consistent within each stratification group, it is recommended to present the results from the main analysis approach (i.e., inverse variance with 0.5 continuity correction) in this table in the main text, and perhaps move the results from the sensitivity analyses to the supplementary material.
Please use the term "not statistically significant" instead of "non-significant" (line 148) or "statistically insignificant" (line 134).
The conclusive statements in the Discussion section seem to be contradictory, particularly in the first paragraph.
It is recommended that the manuscript be reviewed for grammar and flow of the English language.